# Design and Validation of the Instrument for the Measurement of Learning and Performance in Football

**DOI:** 10.3390/ijerph17134629

**Published:** 2020-06-27

**Authors:** Juan M. García-Ceberino, Antonio Antúnez, Sergio J. Ibáñez, Sebastián Feu

**Affiliations:** 1Optimization of Training and Sports Performance Research Group (GOERD), University of Extremadura, 10003 Cáceres, Spain; antunez@unex.es (A.A.); sibanez@unex.es (S.J.I.); sfeu@unex.es (S.F.); 2Faculty of Education, University of Extremadura, 06006 Badajoz, Spain; 3Faculty of Sports Science, University of Extremadura, 10003 Cáceres, Spain

**Keywords:** assessment, decision-making, technical skill, expert judge, Aiken’s *V*, Cronbach’s *α*

## Abstract

The assessment of tactical-technical knowledge of football is essential to develop optimal and integral teaching processes for students. Therefore, the aim of this study was to design and validate an instrument so that teachers, coaches, and researchers can observe and codify both the tactical behaviors and technical skills performed by the students in the game of football. The design and validation of the instrument were carried out in four phases: a) review of the literature and previous instruments; b) design of the Instrument for the Measurement of Learning and Performance in Football (IMLPFoot). It assesses all the offensive and defensive play actions, with and without the ball, as well as their three components (decision-making, technical execution, and final result); c) sample selection of experts (*N* = 12); and d) quantitative (Likert-type scale from 1 to 10) and qualitative assessment of degree the pertinence, unambiguity, and importance of each of the 33 items included in the IMLPFoot. Aiken’s *V* coefficient was used to determine content validity. Likewise, internal consistency was calculated using Cronbach’s *α* coefficient. The results showed demanding levels of validity (*V* ≥ 0.77), internal consistency (*α* = 0.983), inter-rater, and intra-rater reliability. Therefore, it is a valid and reliable instrument that makes possible a complete assessment of football in physical education classes and/or in the sports context (out-of-school football).

## 1. Introduction

Innumerable situations arise in the practice of football that require adaptation on the part of the students to respond to the problems that face them [1,2]. Some of these situations involve spatial-temporal factors, game pace, the strengths and weaknesses of the opponent, the team’s own strengths and weaknesses, etc. [3]. The capacity of the student to adapt to the different situations that arise in the context of the game is called tactical knowledge [4]. So, tactic is the main component of football performance and it is essential to study it to achieve optimal and integral training for young students [5].

During the years from 1960 to 1980, objective tests of technical execution abilities were commonly used for assessments [6]. Indicating that performance in the game depended exclusively on technical execution abilities was a mistake, given that these tests did not measure the strategies the students should apply (decision-making) in real game situations [7,8]. In this respect, a student may be technically talented, but unable to respond to the decision-making demands of play. In contrast, a student may be technically limited, but have the ability to respond to the decision-making demands of play [5].

Decision-making cannot be forgotten when assessing sports actions. Three specific aspects of play should be considered when studying football performance: i) the majority of actions occur when the students are not in direct contact with the ball; ii) technically limited students can play football if they have a reasonable level of tactical understanding; and iii) tactical understanding can imply efficient and/or effective technical execution [9]. Nowadays, observational instruments are commonly used in tactical-technical assessments of play to analyze all the factors and elements that condition the students’ behavior and tactical performance in game situations [10,11], validly and objectively [12].

Regarding sports initiation, different instruments measure behavior and tactical performance in team sports using systematic observation: i) the Game Performance Assessment Instrument (GPAI) [8], that assesses the involvement and performance of each student in the game through tactical understanding and technical execution, and it is valid for both the school and scientific contexts; ii) the Team Sport Assessment Procedure (TSAP) [13], that assesses the interactions between strategic/tactical efficacy and technical efficacy, providing information on the students’ performance when the team is in possession of the ball; and iii) the Procedural Tactical Knowledge Test or KORA [14,15], that assesses the perception of the students to signal their ability to receive a pass and achieve correct orientation, as well as their capacity to recognize spaces. Arias-Estero et al. [16] state that the GPAI is the most commonly used instrument for tactical assessment in all team sports, followed by the TSAP. Both were designed to offer teachers and coaches instruments to assess their students, and they represent a starting point for the design and validation of new assessment instruments for team sports: handball [17], rugby [12], or basketball [10,18,19]. There is also an observational instrument for technical-tactical actions in the individual sport of tennis [20].

Specifically, in the sport of football, there are several instruments that assess decision-making and technical ability: i) the System of Tactical Assessment in Soccer (FUT-SAT) [5,9], that makes it possible to assess the tactical knowledge shown by students in real game situations, based on the ten tactical principles defined in football by da Costa et al. [21] (penetration, offensive coverage, mobility, space, offensive unity, delay, defensive coverage, balance, concentration, and defensive unity). This instrument should not be applied to students under 12–13, as they still have not developed abstract thinking to be able to understand these principles [22]; ii) the Game Performance Evaluation Tool, GPET, in its specific version for football [23], that analyzes the decisions taken by the students in the game based on the tactical problems in attack proposed by Bayer [24] (retaining possession of the ball, advancing towards the opposing goal and achieving a goal) and also analyzes the components of motor ability in the execution. The GPET, in contrast to the GPAI or TSAP, relates decision-making and technical execution ability with the principles for action in the game [5]. Other specific instruments for football are: iii) the Instrument for the Codification and Recording of Play Actions in Football (SOF-1) [25]; iv) Observational System applied to the Offensive Phase in Football (SoccerEye) (Porto, Portugal) [26]; and v) the Observation Instrument for Technical and Tactical Actions of the Offense Phase in Soccer [27], which is different from the rest of the instruments because it takes into account the action by the goalkeeper.

Figure 1 presents the different observational tactical assessment instruments for team sports.

The design of the instruments to assess team sports using observation makes it possible for the assessment of play not to depend exclusively on the subjective judgement of the researcher [7]. Advances in the assessment of performance behaviors in play will help teachers and coaches to draw solid conclusions about their interventions during the teaching of team sports [10].

The instruments mentioned for the tactical-technical assessment of football present some limitations, as they do not take account of all the phases of play (attack and defense), all play abilities (with or without the ball), or the components of play actions (decision-making, technical execution, and final result). Regarding the tactical-technical assessment of basketball, the BALPAI [10] does take all these elements into account. Thus, the objective of this study was to present the procedures used in the design and validation, by a panel of experts, of an instrument that permits the assessment of learning and performance in football: the Instrument for the Measurement of Learning and Performance in Football (IMLPFoot). In the validation, the experts assessed: i) the offensive and defensive play actions; ii) the actions with and without the ball; and iii) the three components of play actions.

## 2. Materials and Methods

This instrumental research [28] aimed to design and validation a functional, valid, and reliable instrument [29], to be subsequently used in the educational and/or sports context (out-of-school football).

### 2.1. Participants

The participants who collaborated in the validation of the IMLPFoot were deliberately and intentionally selected according to the degree established by Rodríguez et al. [30]. Therefore, experts must meet at least 80% of the inclusion criteria established by the researcher to be classified as experts. This procedure was used in previous studies [10,31,32]. Similarly, a panel of experts was sought with a confirmed trajectory in the study subject (football training) and capable of forming judgments and assessments which would help the researcher [33].

Initially, collaboration was requested from 45 experts, who had to fulfil at least 4 of the 5 (80%) criteria to form part of the panel of experts (Table 1). Finally, the sample of participants who provided the information requested to assess the IMLPFoot, correctly and on time, was composed of a group of 12 experts (a 26.66% success rate in participation). The participation of experts specializing in football was insufficient. Most of these experts did not answer, so that participation was requested from experts who were specialists in other team sports (basketball and handball) to be able to achieve an adequate number of experts (≥10) [34,35]. Of the 12 participants, 9 (75%) were experts in football. The experts were intentionally selected since they have knowledge and publications in the field of teaching or training football. None of the experts received any gratuity for their participation, as their contributions were totally voluntary.

The inclusion criteria established for forming part of the panel of experts were: i) to have a Ph.D. in Sports Science (C1); ii) to possess a university degree in the area of Physical Activity and Sport (Sports Science with a specialization in football) and also be a university lecturer in the discipline of football (C2); iii) to have a license as a football coach (C3); iv) to have had 10 years’ experience as a university lecturer and/or football coach (C4); and v) to have publications on teaching or training football (C5).

### 2.2. Content Validity, Internal Consistency, and Objectivity of the IMLPFoot

Hernández et al. [36] indicate that every validation of new instruments should fulfil the following requisites: i) validity; ii) reliability; and iii) objectivity. These requisites should be dealt with together, as if any of them were missing the validation instrument would not be useful to develop an investigation.

*Content validity*. Content validity is defined as the degree to which the selected items adequately represent the measurement instrument [37]. This study used the technique of the validation of experts to determine an optimal level of content validity [38]. The experts evaluated for each of the items that make up the IMLPFoot, the three dimensions defined by Tejada [39]: i) relevance (the adequacy of each item regarding the assessment objectives of the instrument); ii) unambiguity (clarity in the wording of each item, so that the experts understand the same meaning); and iii) importance (the relevance of each item for the assessment objectives of the instrument). These dimensions were evaluated on a Likert-type scale of 1 to 10. The experts were also asked to express their opinions or suggestions for each of the items in order to clarify specific aspects (qualitative evaluation) [12,40,41].

*Internal consistency*. Cronbach’s *α* coefficient [42] was used to measure the internal consistency of the items that compose the IMLPFoot, and to show if all the items in the assessment instrument measure the same and can be added together to give a single total score [43].

*Objectivity*. For Hernández et al. [36] (p.197), the objectivity of an assessment instrument is “the degree to which the instrument is permeable to the biases and tendencies of the researcher who administers, rates and interprets it”. Objectivity in this study was reinforced through the request for collaboration from experts with a recognized trajectory in the study object, as well as the application of the same instructions and conditions for all the participants. Furthermore, inter-rater and intra-rater reliability was calculated to contrast that the validation instrument provides objective data. According to Thomas et al. [37], an instrument cannot be valid if it lacks reliability.

### 2.3. Instrument: IMLPFoot

The IMLPFoot offers Physical Education teachers, coaches, and researchers an instrument for observing and codifying the tactical behavior and technical abilities of young students learning to play football. It was designed to assess six players (+ goalkeepers) when playing small-sided games [44]. Small-sided games make it possible to verify what the students really know and can execute during play [7]. These types of games also allow students to participate more actively, given that in smaller spaces different technical-tactical abilities (controlling, shooting, dribbling, intercepting, clearing, etc.) occur with greater frequency [45]. The play situation analyzed, 3vs3 (+ goalkeepers) (Figure 2), makes the designed and validated assessment instrument functional and useful for the educational and/or sports context, as it makes it possible to assess eight students at the same time.

This instrument includes a total of 11 play actions: seven in attack and four in defense, with and without the ball (Table 2). Three differentiated components/items are assessed for each of the 11 actions: i) decision-making (DM); ii) technical execution (TE); and iii) the final result (FR). In team sports, the DM is the capacity to select the adequate action from a series of alternatives, and TE the ability to carry out the said action in the context of play [46]. These two items will determine the FR of the play action, which constitutes the third item measured by the assessment instrument.

In each of the 11 play actions, each item (DM, TE, and FR) is coded according to the observed adequacy: i) inadequate action; ii) neutral action; and iii) adequate action. This coding proposal differs from the majority of existing instruments, given that they develop two levels of assessment (adequate/inadequate; successful/unsuccessful; appropriate/inappropriate). The IMLPFoot, like the IAD-BB [47] or the BALPAI [10], incorporates a medium level of adequacy, as it is considered that actions are performed in football which although not optimal are not harmful either.

Two procedures are suggested for establishing the adequacy or suitability of each of the items in each of the play actions of the instrument: i) summative; and ii) according to levels.

*The summative procedure*. Two criteria were established for an action to be assessed as adequate. Thus, if the observed action fulfils the two criteria, it is considered an adequate action; if it only fulfils one criterion, it is considered a neutral action; and if it fulfils neither of the criteria, it is considered an inadequate action.

*The levels procedure*. Three levels of adequacy were established for each play action, according to how the action is observed (inadequate, neutral or adequate).

*Coding system and calculation of indicators*. The IMLPFoot follows the coding model proposed by Ibáñez et al. [10] in the BALPAI. Thus, the 11 play actions are codified using a categorical system and awarding a score for each item (DM, TE, and FR): 1 = inadequate action; 2 = neutral action; and 3 = adequate action (Table A1). For each possession of the ball [48], all the play actions (attack or defense, with or without the ball) performed by each student are coded according to the DM, TE, and RF. At the end of the matches, once all the play actions have been coded, the Participation in the Game (PG) indicator is calculated. This indicator is used to calculate the Performance Index (PI) obtained by each student, together with the Decision-Making Performance Index (DM-PI), the Technical Execution Performance Index (TE-PI), the Final Result Performance Index (FR-PI) and the Total Performance Index (Total-PI). Table 3 presents the calculations of these indicators.

All the points awarded to each student for each item of the play action are added together. This figure is an absolute indicator of the performance achieved by a particular student. The PG is also an absolute value and indicates the number of actions in which the student has been involved. Lastly, the PI is a relative value that is obtained by dividing the total number of points awarded to the student by the number of actions in which they have been involved. In this way, it is possible to obtain the PI of each of the play actions or the total PI for the game. This index makes it possible to contrast the performance of two different students or of the same student at two different times. The values for the PI can be between 1 and 3: i) PI = 1, indicates that all the play actions performed have been inadequate; PI = [1–2], indicates that the actions performed have mostly been inadequate; iii) PI = 2, indicates that all the play actions performed have been neutral; PI = [2–3], indicates a predominance of adequate actions; and PI = 3, indicates that the play actions performed have been adequate. These values can be calculated globally or broken down into offensive and defensive actions to determine performance in each of these two phases of play separately.

### 2.4. Procedure

First, a revision of the literature was carried out on assessment instruments of the performance of students in team sports in general and football in particular (Figure 1). This revision served as a starting point, as the design of the IMLPFoot took into account the shortcomings of other instruments, as well as the positive aspects that could be adapted for this new instrument.

After the instrument had been designed, the inclusion criteria were determined for subjects to be considered experts to be able to achieve greater objectivity and quality in the validation. Once the sample of experts had been selected, they were emailed the documentation necessary for validating the instrument: i) a formal institutional presentation of the study: ii) a form on which the experts had to select the criteria they fulfilled; iii) the IMLPFoot; and iv) a template where the experts could quantitatively assess the degree of pertinence, unambiguity and importance of the items (DM, TE, and FR) on a scale from 1 to 10 of all the play actions, as well as carry out a qualitative assessment to improve them.

Once the final number of subjects participating in the panel of experts was known, and their evaluations were obtained, the criteria were determined for the elimination, modification, or acceptance of the items according to the value obtained after calculating Aiken’s *V* coefficient [49]. In this way, the items which attained an optimal score were retained, the items susceptible to improvement were modified according to the suggestions of the experts and the items which received an inadequate score were reformulated [50]. The items with an optimal score were also improved if any suggestions were offered. Then, the index of internal consistency of the items was calculated.

After the content validation process and the incorporation of the modifications suggested by the experts, the final version of the IMLPFoot was defined.

Finally, inter-rater reliability was calculated using the Free-Marginal Multirater Kappa program [51]. For this, fifth grade students were recorded while they were playing a five-minute match. Likewise, three coders (Physical Education teachers and experts in football) received training to learn how to use the IMLPfoot. Then, they assessed all the play actions performed by each student, registering 101 attack actions and 64 defense actions. The values indicated a considerable-almost perfect agreement [52] in the assessed play actions 

Intra-rater reliability of one of the coders was also calculated using Cohen’s Kappa index [53]. The values showed a good and very good agreement in the play actions [52].

### 2.5. Statistical Analyses

The content validity to validation of the IMLPFoot according to the evaluations of the panel of experts was calculated with Aiken’s *V* coefficient [49]. This coefficient makes it possible to quantify the relevance of an item according to the opinion of a group of experts. Its value oscillates between 0 and 1, with the latter figure showing perfect agreement among the experts with regard to the contents evaluated. The score obtained with the calculation of this coefficient established which items should be eliminated, modified or acceptance. The algebraic equation modified by Penfield et al. [54] was used to calculate Aiken’s *V* coefficient.
(1)$$V=\frac{\bar{X}-1}{k}$$

The free Visual Basic 6.0 program [55] was used to obtained three factors: i) the range of evaluations (maximal evaluation-minimal evaluation); ii) Aiken’s *V* coefficient; and iii) the confidence intervals of 90%, 95% and 99%, using the score method [54].

Once Aiken’s *V* coefficient was calculated, it was necessary to establish the criteria for the elimination, modification or acceptance of the items. The exact critical value for the acceptance of Aiken’s *V* was calculated using the initial formula proposed by Aiken [49], applying the central limit theorem for large samples (*m* > 25). The number of experts was 12 (*n*), the number of items 33 (*m*), with a response scale of 10–1 = 9 (*c*) and applying a 95% or 99% confidence level (*z*).
(2)$$V=\frac{z}{0.2\sqrt{\frac{3\mathrm{mn}\left(c\;-\;1\right)}{\left(c+1\right)}}}+0.5$$

A confidence level of 95% was considered to obtain the exact critical value for accepting the items, resulting in a value of 0.77. Similarly, to obtain the cut-off point for the modification of the items a confidence level of 99% was considered, resulting in a value of 0.88. So that, all the items with lower values with a 95% confidence level (*V* < 0.77) were eliminated, the items with values between 95% and 99% (*V* = 0.77 and 0.88) were modified, and lastly the items with a confidence level of 99% (*V* > 0.88) were considered optimal (Table 4). A very demanding criterion was established for the elimination, modification or acceptance of the items, based on the criteria obtained in other evaluation instruments: the BALPAI (*V* = 0.68 and 0.75) [10]; the Observational Instrument for the Technical-Tactical Actions in Singles Tennis (*V* = 0.70 and 0.81) [20]; and the Observation Instrument for Technical and Tactical Actions of the Offense Phase in Soccer (*V* = 0.70 and 0.80) [27]. These instruments also considered the 95% confidence level to obtain the exact critical value for include of the items and the 99% confidence level to obtain the cut-off point for the modification of the items.

The analysis of the internal consistency of the items that make up the IMLPFoot was calculated using Cronbach’s *α* coefficient [42]. The analysis of internal consistency was performed using the statistical SPSS 21.0 program (IBM Corp. Released 2012. IBM SPSS Statistics for Windows, Version 21. Armonk, NY: IBM Corp).

Inter-rater reliability was calculated using the Free-Marginal Multirater Kappa (Multirater *_Kfree_*) program [51]. Likewise, Cohen’s Kappa index [53] was used to determine intra-rater reliability.

## 3. Results

*Content validity*. The results of the instrument validation process are described below. Table 5 and Table 6 present the obtained results after the calculation of Aiken’s *V* coefficient and its confidence intervals of 95% and 99% of the items that make up the validation instrument. The values obtained suggest excellent content validity. It was not necessary to eliminate any item, as they presented values equal to or above 0.77 in the aspects of pertinence and importance (adequacy). The qualitative evaluations of the experts were applied in all the items to clarify aspects and improve the instrument.

In spite of the fact that some items present values lower than 0.77 in the aspect of unambiguity (Table 6), they were not eliminated as this referred to their wording. They were re-written and improved following the suggestions of the experts.

Table 7 presents some of the qualitative evaluations issued by the experts and the action taken to improve the IMLPFoot, as an example.

*Internal consistency*. The values obtained for the internal consistency of the validation instrument using the calculation of Cronbach’s *α* coefficient are shown in Table 8. When considering the reliability of new instruments, values of over 0.90 are considered excellent [56].

*Inter-rater and intra-rater reliability*. Table 9 shows the values obtained of the inter-rater reliability. These values indicate a considerable-almost perfect agreement in the assessed play actions [52].

Intra-rater reliability using Cohen’s Kappa index shows a moderate agreement in the following play action: marking the player without the ball (*K* = 0.60). A good agreement in the following play actions: passing (*K* = 0.70), passing and playing (unmarking) (*K* = 0.79), occupying free spaces by the player without the ball (*K* = 0.62) and marking the player with the ball (*K* = 0.75). Moreover, a very-good agreement in the following play actions: running with the ball (*K* = 1.00), shooting (*K* = 0.86), controlling the ball (*K* = 0.82), rebound taken advantage of by the attackers (*K* = 1.00), assisting and recovering/defensive change (*K* = 1.00) and rebound taken advantage of by the defenders (*K* = 1.00).

## 4. Discussion

In the sports field, in spite of technological advances, it is common to use observational instruments to collect information that makes it possible to analyze and describe the dynamics of the game [11]. The objective of this study was to design and validate the IMLPFoot for specific and general assessment in the game of football. This instrument presents excellent levels of content validity, internal consistency and inter-rater and intra-rater reliability.

The IMLPFoot for football, together, the BALPAI for basketball [10], are the most complete instruments existing in the literature because they evaluate all the offensive and defensive play actions, with and without the ball, as well as their three components (decision-making, technical execution, and final result). The IMLPFoot follows the structure and coding model proposed in the BALPAI.

A panel of experts was responsible for the validation of the instrument, whose suggestions were indispensable in its development [57]. In the case of studies that involve the judgement of experts, a series of recommendations should be taken into account: the quality of the inclusion criteria, the number of experts necessary, the preparation of the instructions and evaluation templates, the procedure for collecting the quantitative and qualitative statistics, as well as a suitable statistical analysis to give validity and reliability to the new instrument [33,40]. These recommendations are similar to those used in this study, as well as in previous studies related to the design and validation of instruments [12,41,50]. According to Paixão et al. [58], both the design and validation should show strong scientific rigor.

The number of experts who participated offering their evaluation of all the items making up the instrument was 26.66% (12 experts) of the initially detected population according to the demanding inclusion criteria [30] and complying with the requisites determined in the literature. Thus, the sample of experts necessary for the validation of instrument was adequate according to Robles et al. [34], who consider that ten or more experts is a reliable sample size. Similarly, Rubio et al. [35] determinate that between two and twenty experts are necessary. In this study, it was necessary to fit the profile of the panel of experts to the subject of study, as mentioned by Juan-Llamas [59].

The experts quantitatively evaluated each of the items in the IMLPFoot on a Likert-type scale of 1 to 10. Different studies use this same range in the validation of new instruments [10,20,60]. However, other studies use other different scales: from 1 to 5 [19,32,61] or 1 to 6 [58]. The qualitative evaluations of the experts were of great importance in the development and improvement of the items in the instrument. Most of them were aimed at improving the wording and clarifying the concepts to avoid uncertainty in the future coders. This made it possible to define the items in the instrument in a clearer and more precise manner, producing a significant improvement [62].

Content validity was calculated using Aiken’s *V* coefficient [49]. A confidence interval of 95% was established for the elimination or acceptance of an item, and of 99% for its modification. In this study, the values of Aiken’s *V* for all the items were equal to or above 0.77 in the aspects of pertinence and importance (adequacy), and therefore no item had to be eliminated. Regarding the minimal values proposed in the literature (*V* = 0.70) [54], the IMLPFoot attained very demanding levels of content validity. All the items, even the ones that obtained excellent levels of content validity, were improved following the suggestions of the experts. Several studies used this coefficient to determine the elimination or acceptance of an item in the validation of instruments [27,58,63].

Similarly, the internal consistency of the IMLPFoot was calculated using Cronbach’s *α* coefficient [42], attaining a value of 0.983. In this regard, Field [56] considers values of over 0.90 to be excellent. The internal consistency obtained by the IMLPFoot was higher than that of other instruments: the Observational Instrument to Basketball Referee Evaluation (*α* = 0.712) [31]; the Coaches’ Training Profile Questionnaire (*α* = 0.890) [64]; the Observational Instrument to know Competitive Performance Indicators in Football a-side-5 for blind people (*α* = 0.894) [32]; the Questionnaire to Assess Football Initiation Coaches’ Training (*α* = 0.915) [58]; or two intervention programs for teaching basketball in the school context (*α* = 0.960) [65].

Possessing excellent knowledge and specific experiences will help students to make the correct decisions and successfully resolve situations of different levels of uncertainty [4]. The IMLPFoot was designed so that teachers, coaches, and researchers could make easy and objective observations. In this respect, inter-rater reliability was calculated using the Free-Marginal Multirater Kappa program [51]. Three coders evaluated a football match. The values indicated a considerable-almost perfect agreement in the assessed play actions [52]. Different studies also analyzed the agreement among observers [10,19,27] after designing new instruments. Likewise, intra-rater reliability using Cohen’s Kappa index provided good and very-good agreement in the play actions [52,53].

The development of the IMLPFoot provide the numerous observational studies that are carried out on football a valid and reliable instrument to collect the data. The panel of experts that participated in the validation of the instrument showed an excellent understanding of the items and definitions. These types of instruments are necessary, as tactical knowledge can be developed and learned, and should be progressively assessed during students’ training.

Among the limitations, it should be pointed out that the IMLPFoot has many assessment elements; and therefore, the coders are recommended to undergo a teaching-learning process to ascertain how it should be implemented before using it. This process will lead to a better use and recognition of play behaviors and is aimed at avoiding the uncertainty that the instrument may generate.

## 5. Conclusions

The values obtained in the validation process highlight that the IMLPFoot possesses a high degree of content validity, internal consistency, and reliability, and it is thus valid and reliable for measuring the learning and performance of students of football in Physical Education classes and/or the sports context. The strong point of the IMLPFoot, in comparison with the rest of the tactical-technical assessment instruments for football, is that it assesses all the offensive and defensive play actions, with and without the ball, as well as their three components (decision-making, technical execution, and final result) performed by students of football in the game. It also makes it possible to resolve specific research problems.

## Figures and Tables

**Figure 1 ijerph-17-04629-f001:**
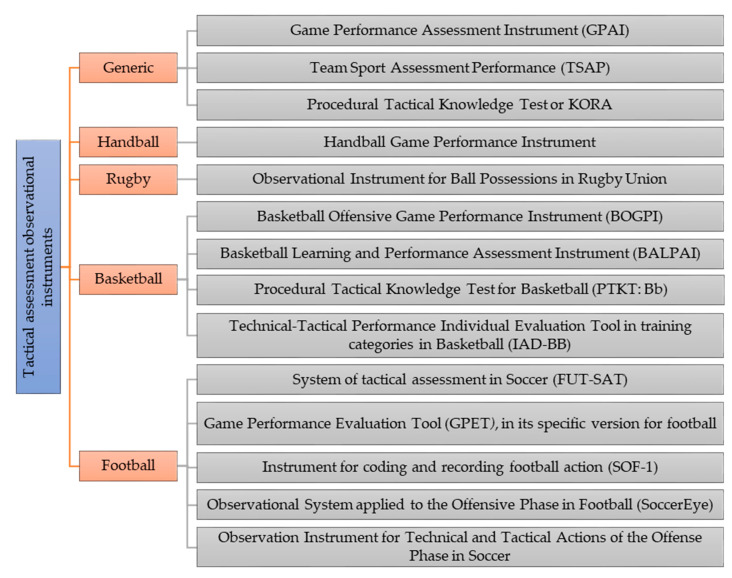
Observational instruments for tactical assessment in team sports.

**Figure 2 ijerph-17-04629-f002:**
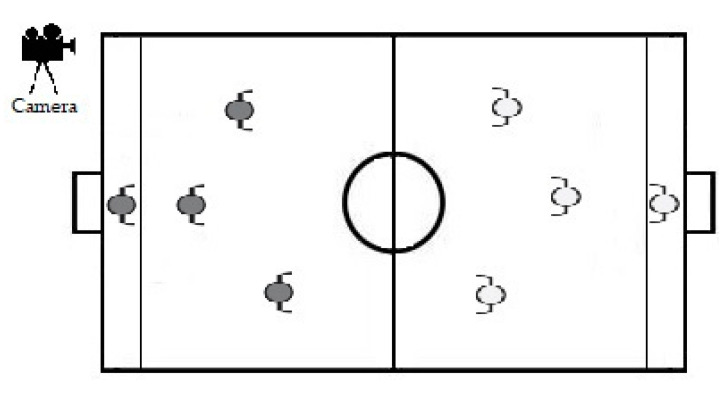
Play situation 3vs3 (+ goalkeepers).

**Table 1 ijerph-17-04629-t001:** Inclusion criteria fulfilled by the experts.

Criteria	E1	E2	E3	E4	E5	E6	E7	E8	E9	E10	E11	E12
**C1**	X	X	X	X	X	X	X	X	X	X	X	X
**C2**	X	X	X		X	BB	BB	X	HB		X	X
**C3**		UPro		UPro				UA		UPro	UPro	
**C4**	X	X	X	X	X	X	X	X	X	X	X	X
**C5**	X	X	X	X	X	X	X	X	X	X	X	X
***%***	80	100	80	80	80	80	80	100	80	80	100	80

Note: *E* = expert; *C* = inclusion criterion; BB = basketball; HB = handball; UPro = UEFA Pro license; UA = UEFA A license.

**Table 2 ijerph-17-04629-t002:** Play actions in the Instrument for the Measurement of Learning and Performance in Football (IMLPFoot).

Attacking Play Actions (7)	Defending Play Actions (4)
Running with the ball	Marking the player with the ball
Shooting	Marking the player without the ball
Passing	Assisting and recovering/defensive change
Controlling the ball	Rebound taken advantage of by the defenders
Passing and playing (unmarking)	
Occupying free spaces by the player without the ball	
Rebound taken advantage of by the attackers	

**Table 3 ijerph-17-04629-t003:** Calculation of the game performance (GP) and PI indicators.

Indicators	DM	TE	FR
**Student A**	Sum of DM points	Sum of TE points	Sum of FR points
**PG**	PG = Number of total actions performed by student A		
**PI**	DM-PI= DM Pts/PG	TE-PI = TE Pts/PG	FR-PI = FR Pts/PG
	Total PI = (DM-PI + TE-PI + FR-PI)/3		

Note: *DM* = decision-making; *TE* = technical execution; *FR* = final result; *PG* = participation in the game; *PI* = performance index; *DM-PI* = decision-making performance index; *TE-PI* = technical execution performance index; *FR-PI* = final result performance index; *Pts* = points.

**Table 4 ijerph-17-04629-t004:** Criteria for the elimination, modification or acceptance of the items.

Aspect	Criteria	Unambiguity		
		> 0.88	[0.77 – 0.88]	< 0.77
**Pertinence + Importance** **(Adequacy)**	> 0.88	Correct	U modified	U modified
	[0.77 – 0.88]	A modified	A and U modified	A and U modified
	< 0.77	Eliminated	Eliminated	Eliminated

Note: *A* = adequacy; *U* = unambiguity.

**Table 5 ijerph-17-04629-t005:** Indexes of content validity (Aiken’s *V*) and CI of 95% and 99% in the aspects of pertinence and importance.

	Pertinence						Importance					
			95% CI		99% CI				95% CI		99% CI	
Item	*M ± SD*	*V*	*Low.*	*Upp.*	*Low.*	*Upp.*	*M ± SD*	*V*	*Low.*	*Upp.*	*Low.*	*Upp.*
1	8.92 ± 2.23	0.88 *	0.80	0.93	0.78	0.94	9.25 ± 0.97	0.92	0.85	0.95	0.82	0.96
2	9.50 ± 0.80	0.94	0.88	0.97	0.86	0.98	9.33 ± 0.89	0.93	0.86	0.96	0.83	0.97
3	8.92 ± 2.57	0.88 *	0.80	0.93	0.78	0.94	8.83 ± 2.55	0.87 *	0.79	0.92	0.77	0.93
4	9.17 ± 1.47	0.91	0.84	0.95	0.81	0.96	9.58 ± 0.67	0.95	0.90	0.98	0.87	0.98
5	9.42 ± 0.79	0.94	0.87	0.97	0.85	0.97	9.42 ± 0.79	0.94	0.87	0.97	0.85	0.97
6	8.08 ± 2.71	0.79 *	0.70	0.85	0.67	0.87	8.08 ± 2.71	0.79 *	0.70	0.85	0.67	0.87
7	8.92 ± 1.56	0.88 *	0.80	0.93	0.78	0.94	8.92 ± 1.51	0.88 *	0.80	0.93	0.78	0.94
8	9.00 ± 1.48	0.89	0.82	0.93	0.79	0.94	9.08 ± 1.51	0.90	0.83	0.94	0.80	0.95
9	8.83 ± 1.70	0.87 *	0.79	0.92	0.77	0.93	8.83 ± 1.70	0.87 *	0.79	0.92	0.77	0.93
10	9.17 ± 1.03	0.91	0.84	0.95	0.81	0.96	8.67 ± 1.67	0.85 *	0.77	0.91	0.74	0.92
11	9.25 ± 1.22	0.92	0.85	0.95	0.82	0.96	9.25 ± 1.22	0.92	0.85	0.95	0.82	0.96
12	9.08 ± 1.24	0.90	0.83	0.94	0.80	0.95	8.75 ± 1.66	0.86 *	0.78	0.91	0.76	0.92
13	9.25 ± 0.87	0.92	0.85	0.95	0.82	0.96	9.25 ± 0.87	0.92	0.85	0.95	0.82	0.96
14	8.25 ± 2.34	0.81 *	0.72	0.87	0.69	0.88	8.08 ± 2.47	0.79 *	0.70	0.85	0.67	0.87
15	8.83 ± 1.47	0.87 *	0.79	0.92	0.77	0.93	8.75 ± 1.54	0.86 *	0.78	0.91	0.76	0.92
16	9.25 ± 0.87	0.92	0.85	0.95	0.82	0.96	9.33 ± 0.78	0.93	0.86	0.96	0.83	0.97
17	8.58 ± 1.78	0.84 *	0.76	0.90	0.73	0.91	8.58 ± 1.78	0.84 *	0.76	0.90	0.73	0.91
18	9.33 ± 0.78	0.93	0.86	0.96	0.83	0.97	9.25 ± 0.97	0.92	0.85	0.95	0.82	0.96
19	8.36 ± 2.58	0.82 *	0.73	0.88	0.71	0.89	8.18 ± 2.52	0.80 *	0.71	0.86	0.68	0.88
20	8.27 ± 2.53	0.81 *	0.72	0.87	0.69	0.89	8.27 ± 2.53	0.81 *	0.72	0.87	0.69	0.89
21	8.36 ± 2.58	0.82 *	0.73	0.88	0.70	0.89	8.27 ± 2.53	0.81 *	0.72	0.87	0.69	0.89
22	8.18 ± 2.52	0.80 *	0.71	0.86	0.68	0.88	8.18 ± 2.52	0.80 *	0.71	0.86	0.68	0.88
23	8.09 ± 2.55	0.79 *	0.70	0.85	0.67	0.87	8.00 ± 2.83	0.78 *	0.69	0.85	0.66	0.86
24	8.27 ± 2.53	0.81 *	0.72	0.87	0.69	0.89	8.27 ± 2.53	0.81 *	0.72	0.87	0.69	0.89
25	9.33 ± 0.78	0.93	0.86	0.96	0.83	0.97	9.33 ± 0.78	0.93	0.86	0.96	0.83	0.97
26	8.67 ± 1.97	0.85 *	0.77	0.91	0.74	0.92	8.83 ± 1.99	0.87 *	0.79	0.92	0.77	0.93
27	9.33 ± 0.78	0.93	0.86	0.96	0.83	0.97	9.33 ± 0.78	0.93	0.86	0.96	0.83	0.97
28	9.17 ± 1.03	0.91	0.84	0.95	0.81	0.96	9.25 ± 0.97	0.92	0.85	0.95	0.82	0.96
29	8.58 ± 1.93	0.84 *	0.76	0.90	0.73	0.91	8.33 ± 2.19	0.81 *	0.73	0.88	0.70	0.89
30	8.83 ± 1.47	0.87 *	0.79	0.92	0.77	0.93	8.75 ± 1.71	0.86 *	0.78	0.91	0.76	0.92
31	9.36 ± 1.03	0.93	0.86	0.96	0.84	0.97	9.27 ± 1.01	0.92	0.85	0.96	0.83	0.96
32	8.45 ± 2.21	0.83 *	0.75	0.89	0.72	0.90	8.45 ± 2.34	0.83 *	0.75	0.89	0.72	0.90
33	9.36 ± 0.81	0.93	0.86	0.96	0.84	0.97	9.36 ± 0.81	0.93	0.86	0.96	0.84	0.97

Note: *M* = mean; *SD* = standard deviation; *V* = Aiken’s *V*; CI = confidence interval; *Low.* = lower limit; *Upp.* = upper limit. * *V* = [0.77 – 0.88].

**Table 6 ijerph-17-04629-t006:** Indexes of content validity (Aiken’s *V*) and CI of 95% and 99% in the aspect of unambiguity.

Item	Unambiguity					
	*M ± SD*	*V*	95% CI		99% CI	
			*Low.*	*Upp.*	*Low.*	*Upp.*
1	7.67 ± 2.64	0.74 ^†^	0.65	0.81	0.62	0.83
2	9.08 ± 1.51	0.90	0.83	0.94	0.80	0.95
3	8.58 ± 1.98	0.84 *	0.76	0.90	0.73	0.91
4	8.00 ± 1.86	0.78 *	0.69	0.85	0.66	0.86
5	8.67 ± 1.56	0.85 *	0.77	0.91	0.74	0.92
6	7.92 ± 2.11	0.77 *	0.68	0.84	0.65	0.85
7	8.25 ± 1.82	0.81 *	0.72	0.87	0.69	0.88
8	8.00 ± 2.17	0.78 *	0.69	0.85	0.66	0.86
9	8.33 ± 2.02	0.81 *	0.73	0.87	0.70	0.89
10	8.75 ± 1.71	0.86 *	0.78	0.91	0.76	0.92
11	8.17 ± 1.99	0.80 *	0.71	0.86	0.68	0.88
12	8.83 ± 1.70	0.87 *	0.79	0.92	0.77	0.93
13	8.37 ± 1.30	0.85 *	0.77	0.91	0.74	0.92
14	8.33 ± 1.83	0.81 *	0.73	0.88	0.70	0.89
15	7.92 ± 2.31	0.77 *	0.68	0.84	0.65	0.85
16	8.25 ± 1.71	0.81 *	0.72	0.87	0.69	0.88
17	8.33 ± 1.92	0.81 *	0.73	0.88	0.70	0.89
18	8.17 ± 2.12	0.80 *	0.71	0.86	0.68	0.88
19	7.55 ± 2.58	0.73 ^†^	0.64	0.80	0.61	0.82
20	7.91 ± 2.81	0.77 *	0.68	0.84	0.65	0.85
21	7.91 ± 2.07	0.77 *	0.68	0.84	0.65	0.85
22	7.91 ± 2.43	0.77 *	0.68	0.84	0.65	0.85
23	7.91 ± 2.74	0.77 *	0.68	0.84	0.65	0.85
24	8.00 ± 2.05	0.78 *	0.69	0.85	0.66	0.86
25	7.33 ± 2.35	0.70 ^†^	0.61	0.78	0.58	0.80
26	8.08 ± 1.93	0.79 *	0.70	0.85	0.67	0.87
27	8.67 ± 1.78	0.85 *	0.77	0.91	0.74	0.92
28	8.00 ± 2.45	0.78 *	0.69	0.85	0.66	0.86
29	8.50 ± 1.73	0.83 *	0.75	0.89	0.72	0.90
30	8.83 ± 1.59	0.87 *	0.79	0.92	0.77	0.93
31	7.91 ± 2.26	0.77 *	0.68	0.84	0.65	0.85
32	8.27 ± 2.00	0.81 *	0.72	0.87	0.69	0.89
33	9.18 ± 0.98	0.91	0.84	0.95	0.81	0.96

Note: *M* = mean; *SD* = standard deviation; *V* = Aiken’s *V*; CI = confidence interval; *Low*. = lower limit; *Upp*. = upper limit. **V* = [0.77 – 0.88]; ^†^*V* < 0.77.

**Table 7 ijerph-17-04629-t007:** Qualitative evaluations issued by the experts.

Item	Qualitative Evaluation	Action
1	E1: *“Further clarify tactical intentionality”*	A clearer definition was determined for tactical intentionality
4	E9: *“Change the concept of deliver for kick or shot”*	The concept of deliver was changed for kick or shot
9	E4: *“Include the concept of the instant or moment of the pass”*	The concept of the instant or moment of the pass was included
15	E2: *“Replace «is allowed to get free» with «gets free»”*	«Is allowed to get free» was replaced with «gets free»
17	E4: *“The concept of situation, position, and orientation should be included”*	The concepts of situation, position, and orientation were included
25	E5: *“Specify the distance with a range of what is considered far or near to avoid subjectivity”*	Distance was specified with a range to avoid subjectivity

Note: *E* = expert.

**Table 8 ijerph-17-04629-t008:** Internal consistency of the items that make up the IMLPFoot.

Statistic	Pertinence	Importance	Unambiguity	Total
***n***	33	33	33	99
***α***	0.951	0.955	0.952	0.983

**Table 9 ijerph-17-04629-t009:** Inter-rater reliability values and percent overall agreement.

Play Actions	*M_Kfree_* DM / %		*M_Kfree_* TE / %		*M_Kfree_* FR / %	
Running with the ball	1.00	100.00	0.87	91.67	1.00	100.00
Shooting	0.83	88.89	0.67	77.78	1.00	100.00
Passing	0.91	94.20	0.96	97.10	1.00	100.00
Controlling the ball	0.93	95.56	0.93	95.56	1.00	10.000
Passing and playing (unmarking)	0.82	87.88	0.82	87.88	0.82	87.88
Occupying free spaces	0.94	95.83	0.94	95.83	0.91	93.75
Rebound (by the attackers)	1.00	100.00	1.00	100.00	1.00	100.00
Marking the player with the ball	0.83	88.89	0.78	85.19	1.00	100.00
Marking the player without the ball	0.82	88.24	0.82	88.24	0.91	94.12
Assisting/defensive change	1.00	100.00	1.00	100.00	1.00	100.00
Rebound (by the defenders)	1.00	100.00	1.00	100.00	1.00	100.00

Note: *DM* = decision-making; *TE* = technical execution; *FR* = final result.

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
