# Peer review of "Design and Validation of the Instrument for the Measurement of Learning and Performance in Football"

_ijerph, 2020, doi:10.3390/ijerph17134629_

Round 1
Reviewer 1 Report
First of all, I would like to thanks the author for the point by point answers and for the improvement of the manuscript. For me is accepted for publication in the current form.
Author Response
Dear reviewer,
I appreciate the acceptance of the article.
Kind regards!
Reviewer 2 Report
The limitations are seriously pointed out by the authors. But I have many doubts about the practical applicability of the instrument, which is a relevant indicator of any evaluation instrument. In addition, although recognized by the authors, it is a serious failure that the inter-rater reliability have not been calculate.
Author Response
Dear reviewer,
First of all, we would like to express our gratitude to reviewer 2 for the time in reviewing our manuscript again and for providing us comments helpful to improve this manuscript quality.
All corrections were marked in red.
--------------------------------
Reviewer’ note: The limitations are seriously pointed out by the authors. But I have many doubts about the practical applicability of the instrument, which is a relevant indicator of any evaluation instrument. In addition, although recognized by the authors, it is a serious failure that the inter-rater reliability have not been calculate.
Authors’ response: Inter-rater reliability was calculated (Line 232 to 234; Line 270 to 271; Line 349 to 352). Excellent reliability values were obtained (see Table 9).
Kind regards!
Reviewer 3 Report
Dear Authors,
In resubmission, you have tried to address some of the highlighted issues, but the expert selection part is still problematic, which still undermines the results.
Line 108 - You still did not manage to explain the inclusion criteria.
Line 110 – How did you come to the 80% inclusion criteria (IC)? Referenced studies are from co-authors papers and in these papers, the inclusion criteria are not backed up by any research (the same as here). They are just set. So 80% IC are backed up by research that does not back up their inclusion criteria? Also in this research, there is no statement of at least 80% of inclusion criteria (in some it is 3 of 4, 4 of 6 – which is not 80%, etc.)
Line 120 - Still, 7 of 12 experts do not have football qualifications, and this study is about football? There is still no report on the level of football qualification of those that had it -was it Uefa C, Uefa B, Uefa A, PRO? That plays an essential part in their “expertise”.
Lines 126 to 130 - Chosen experts are from sports that have different dynamics, play on smaller fields/play with arms, a smaller number of players which impact player movement dynamics, etc. There is no additional reference provided of a study where there was a validation of a specific tool for a specific sport, and there were experts from other sport disciplines (do back up your decision)? A statement that additional experts have publications in coaching or training football does not prove anything as they could contribute to those papers is statistical analysis. So someone can have ten papers in a field of football where they did the statistical analysis – does that bake them a football expert? No
What about inter-rater reliability – From the statement in the limitations part, it is clear that this has not been done. Therefore we do not know if the new measuring instrument yields consistent results, making it unusable in practice until this is done. Authors try to back up this by referencing the work where this has been done after the validation – but all of the referenced work is done by co-authors of this paper where no reliability was performed. This shows that this is not a standard practice in other research, just in the author's work, which shows methodological issues.
Validity and reliability are two critical factors to consider when developing and testing any instrument (e.g., content assessment test, questionnaire) for use in a study. They should, therefore, be done together and not afterward.
Kind regards
Author Response
Dear reviewer 3,
First of all, we would like to express our gratitude to reviewer 3 for the time in reviewing our manuscript again and for providing us comments helpful to improve this manuscript quality. We have found suggestions very constructive and have answered their concerns point by point.
All corrections were marked in red.
--------------------------------
Reviewer’ note: Line 108 - You still did not manage to explain the inclusion criteria.
Authors’ response: Rodríguez et al. [31] do not define any inclusion criteria. They propose that of the inclusion criteria established by the researcher himself, at least 80% of them be met.
--------------------------------
Reviewer’ note: Line 110 – How did you come to the 80% inclusion criteria (IC)? Referenced studies are from co-authors papers and in these papers, the inclusion criteria are not backed up by any research (the same as here). They are just set. So 80% IC are backed up by research that does not back up their inclusion criteria? Also in this research, there is no statement of at least 80% of inclusion criteria (in some it is 3 of 4, 4 of 6 – which is not 80%, etc.).
Authors’ response: The inclusion criteria to be considered an expert were defined by the researches himself. The experts had to fulfil at least 4 of the 5 (80%) inclusion criteria to form part of the panel of experts (Table 1). It was asked by email how many criteria they met out of the 5 (Line 218 to 219).
--------------------------------
Reviewer’ note: Line 120 - Still, 7 of 12 experts do not have football qualifications, and this study is about football? There is still no report on the level of football qualification of those that had it -was it Uefa C, Uefa B, Uefa A, PRO? That plays an essential part in their “expertise.
Authors’ response: Criterion 2 were experts specialized in football during their university education and who currently teach football classes at the university (On line 122). These do not have a license as a football coach, but they have football qualifications. Some experts met both inclusion criteria. Nine (75%) were experts in football.
Table 1 indicated the level of license as a football coach of the experts who met this inclusion criterion.
--------------------------------
Reviewer’ note: Lines 126 to 130 - Chosen experts are from sports that have different dynamics, play on smaller fields/play with arms, a smaller number of players which impact player movement dynamics, etc. There is no additional reference provided of a study where there was a validation of a specific tool for a specific sport, and there were experts from other sport disciplines (do back up your decision)? A statement that additional experts have publications in coaching or training football does not prove anything as they could contribute to those papers is statistical analysis. So someone can have ten papers in a field of football where they did the statistical analysis – does that bake them a football expert? No
Authors’ response: These three experts were intentionally selected because they were knowledgeable about the dynamics of the game of footabll. Selecting one more expert, we would have had 10 experts (minimum required in the literature). But we consider that their knowledge in football would be of great help in the development of the instrument.
--------------------------------
Reviewer’ note: What about inter-rater reliability – From the statement in the limitations part, it is clear that this has not been done. Therefore we do not know if the new measuring instrument yields consistent results, making it unusable in practice until this is done. Authors try to back up this by referencing the work where this has been done after the validation – but all of the referenced work is done by co-authors of this paper where no reliability was performed. This shows that this is not a standard practice in other research, just in the author's work, which shows methodological issues.
Authors’ response: Inter-rater reliability was calculated (Line 232 to 234; Line 270 to 271; Line 349 to 352). Excellent reliability values were obtained (Table9).
Kind regards!
Reviewer 4 Report
Firstly, congratulations to the authors on this great study.
My main comments are concerning the methods, especially the procedure and reliability sub-sections. Before that, I suggest checking the two paragraphs in lines 113-119 and 126-130. The information is similar (is partially repeated) and these two paragraphs could be easily merged. Finally, in lines 232 onwards (reliability), please add more details. How many actions were observed in five minutes, which statistical technique(s) was/were used, and which values in intra- and inter-observer were recorded.
Author Response
Dear reviewer 4,
First of all, we would like to express our gratitude to reviewer 4 for the time in reviewing our manuscript and for providing us comments helpful to improve this manuscript quality. We have found suggestions very constructive and have answered their concerns point by point. All corrections were marked in red.
-------------
Reviewer’ note: I suggest checking the two paragraphs in lines 113-119 and 126-130. The information is similar (is partially repeated) and these two paragraphs could be easily merged.
Authors’ response: Both paragraphs were regrouped into a single paragraph (Line 113 to 122).
-------------
Reviewer’ note: in lines 232 onwards (reliability), please add more details. How many actions were observed in five minutes, which statistical technique(s) was/were used, and which values in intra- and inter-observer were recorded.
Authors’ response: More details were added according to the inter-rater and intra-rater reliability, such as the total number of play actions analyzed, statistical techniques and values obtained.
Kinds regards!
Round 2
Reviewer 2 Report
The authors improved the methodological description and explained my concerns. Therefore, I recommend this article to be published.
Author Response
Dear reviewer 2,
We would like to express our gratitude to reviewer 2 for the time in reviewing our manuscript and accepting it for publication.
Kinds regards!
Reviewer 3 Report
Dear Authors,
I don't know why I have received this paper for the 3rd time as I already rejected this paper twice.
I see you are patching things along the way as they appear, which shows that the study was not well planned.
You added the coaching licenses in the wrong section - They should be in the C2 and not C3 criteria. Also, you reported in the C3 Uefa Pro license to evaluators E4 and E10 that according to the C2 criteria don't have a coaching qualification. What is it then? All of this doesn't make any sense.
(C3) criteria is confusing - to have had 10 years’ experience as a
university lecturer and/or football coach. You can't state and/or (totally different meaning). What is it then? So someone could have only lecturing experience and no football coaching experience to be a football expert?
In the inter-rater reliability, you reported - Three coders (Physical Education teachers and experts in football) received training to learn how to use the IMLPfoot. So this was not done with the data of 12 experts that carried out the validation of the instrument? When was this done? Was it done on the same sample? Intra-rater reliability was done only on 1 evaluator which is not enough. You also don't report in what time frame did he perform the second evaluation. You also didn't report standard error of measurement and average relative error.
Also, table 1 has changed, but you didn't mention any change in expert selection. In table one from version 1 to version 2 a noticeable change in C2 and C3 is noted. In the first version experts E3, E4, E5, and E11 didn't have the coaching license but in the latest version, only the E4 and E10 don't have a football coaching license. Also, the C3 criteria are amended. So did you change the experts or did they retroactively gained experience or coaching licenses?
All of this raises even greater doubts in the execution of your research and therefore, I have to reject your paper again.
Kind regards
Author Response
Dear reviewer 3,
We would like to express our gratitude to reviewer 3 for the time in reviewing our manuscript.
Kind regards!
This manuscript is a resubmission of an earlier submission. The following is a list of the peer review reports and author responses from that submission.
Round 1
Reviewer 1 Report
This research aims to develop and validate an instrument to observe and codify the tactical actions and technical skills performed by the students in football.
From my point of view the main purpose is achieved and the instrument have interest for football training in school and training young athletes. The article is very interesting and is generally elucidative, however, in other to be published, in my opinion, there are some changes that must be performed.
- In general the citation are founded in Spanish language, my suggestion is to at least in line 31 after “Innumerable situations arise in the practice of football that require adaptation on the part of the students to respond to the problems that face them” to be added another more robust citation;
- Point 2.1 Design, is confuse and information that must be in this section are in another topics Examples: line 108 to 121 and line 210 to 217… - In the design you may describe procedures, techniques, aids, or tools for designing the instrument. Suggestion authors may eliminate point 2.1 and perform a small introduction before the point Participants with the content that is in the design; or maintain the design section and transfer all the contents that are in the point participants, instruments and procedures that are direct connected with the design to point 2.1
- Line 251 to 252 “A very demanding criterion was established for the elimination, modification or acceptance of the items.” Please explain the main purpose of these sentence in statistical analyses and why after the confidence level explanation
- Discussion must be improved because have information that are more adequate to the introduction Examples: line 279 to 276; line 288 to 294; line 296 to 297. Please reformulate these sentences to discuss the results and consistence of the procedures instead of presenting information
- Line 275 to 276 “The objective of this study was to design and validate, with a panel of experts, an instrument for specific and general assessment in the game of football (IMLPFoot).” Suggestion: The objective of this study was to design and validate, the IMLPFoot for specific and general assessment in the game of football.
- Line 278 – after “they are the most complete instruments existing in the literature.” add more information to reinforce and justify the observation like one or two main reasons, like you have in the conclusion.
- Line 312 – instead of just “other instruments” and informing the citation, combine with the name of the instrument to better understand what is the comparation
- Line 316 to 319 please improve the sentence “The development of the instrument aimed at providing the numerous observational studies that are carried out on football with a valid and reliable instrument that make it possible to suitably collect the data.” to be more like an implication of this research
- Line 327 to 329 please improve the sentence “In this respect, it is necessary to carry out a pilot test to calculate inter and intra-rater reliability. Different studies analyzed the agreement among observers [9,18,27] after designing new ” to be more like future directions/research
Reviewer 2 Report
Dear Authors,
Overall your study is well carried (procedures and statistical analysis) out however, I see a big problem in your expert selection which undermines your results.
Line 108 - This source is in Spanish and not freely available for me to check the ''so-called inclusion criteria''.
Write down the inclusion criteria as they are from a Spanish book from 1996 (and I suppose readers don't have free access to it) or cite some recent papers (in English) that used these inclusion criteria.
Line 115 - The source of this citation 32 is (Ortega, Jimenez, Palao, and Sainz, 2008) in the source you have written. Cite correctly! Are there any other studies that could support your claim about the sufficient number of experts.
Line 120 - So 7 of your 12 experts don't have football qualifications and this study is about football? And those experts without football qualifications are lecturing in the discipline of football on the Universities? How can this be? Elaborate how can they be good experts in comparison to someone that has more than 10 years of coaching experience and a UEFA PRO coaching license (you don't need a Phd to be an expert or to be a university lecturer) - Also what was the level of qualification of those that had it -was it Uefa C, Uefa B, Uefa A, PRO ??? That plays an important part in their ''expertise''.
Lines 122 to 125 - You chose experts from sports that have a totally different dynamics, play on smaller fields/play with arms, a smaller number of players which impacts player movement dynamics, etc.
Show me one study where there was a validation of a specific tool for a specific sport and there were experts from other sport disciplines?
Line 142 - What about inter-rater reliability - did your experts check your new instrument in two separate occasions? Especially with your expert sample. This should be added.
Discussion and comparison to other research-measurement tools that include similar items is missing.
Because of your selection of the expert group, I have high doubts about the relevancy of your results and in your measurement tool.
Kind regards
Reviewer 3 Report
The authors have submitted a manuscript in which the objective is the design and validation of a new instrument to evaluate learning and performance in football. However, there is a selection bias in the selection of experts and there is no validation of the instrument. In addition, the instrument has practical limitations that have not been described in the limitations.
Title: Design and validation of the Instrument for the Measurement of Learning and Performance in Football
Abstract: the conclusions should be in line with the objectives. However, the authors have concluded with practical applications of their study. In addition, the authors must show results for each objective. The structure provided by the authors is difficult to read.
The authors should deepen the need for a new instrument.
Design: the authors have described the objective of the study and not the design. The authors must rewrite and adjust the content.
The authors have made an error of selection bias. Approximately 59 of the experts did not meet criteria 3 and 2. Besides, the C3 includes two different capacities. An expert may have 10 years of experience at university and not be a football coach.
Construct validity and criterion validity have not been evaluated. These types of validity together with the content validity are necessary to validate an instrument.
How many points do you need to accumulate to consider an appropriate action?
How did your tool prove to be able to differentiate between 1 = inadequate action; 2 = neutral action; and 3 = adequate action?
How many students can you value in a single action? It has not been described
I think that the tool simultaneously values so many aspects that the person who completes the instrument must be very trained. Have you considered this point?
The authors have designed an instrument. However, they cannot assume that they have validated the instrument as some aspects have not been tested.